# Reducing Computation in Recurrent Networks by Selectively Updating State Neurons

## Abstract

Recurrent Neural Networks (RNNs) are the state-of-the-art approach to sequential learning. However, standard RNNs use the same amount of computation at each timestep, regardless of the input data. As a result, even for high-dimensional hidden states, all dimensions are updated at each timestep no matter the choice of recurrent memory cell. Reducing this rigid assumption could allow for models with large hidden states to perform inference more quickly. Intuitively, not all hidden state dimensions need to be recomputed from scratch at each timestep. Thus, recent methods have begun studying this problem by imposing mainly a priori-determined patterns for updating the states at each step. In contrast, we now design a fully-learned approach, SA-RNN, that augments any RNN by predicting discrete update patterns at the fine granularity of independent hidden state dimensions. This is achieved through the parameterization of a distribution of update-likelihoods driven by the input data. Notably, through this approach we impose no assumptions on the structure of the update pattern. Better yet, our method adapts the update patterns online, allowing different dimensions to be updated conditional to the input. To learn which dimensions to update, the model solves a multi-objective optimization problem, maximizing task performance while minimizing the number of updates based on a unified control. Using publicly-available datasets we demonstrate that our method consistently achieves higher accuracy with fewer updates compared to state-of-the-art alternatives. Additionally, our method can be directly applied to a wide variety of models containing RNN architectures.

## 1 Introduction

Recurrent Neural Networks (RNN) are the state-of-the-art approach to many sequential learning problems including speech recognition (Graves et al., 2013), machine translation (Bahdanau et al., 2015), and sequence generation (Graves, 2013; Xu et al., 2015). However, RNNs typically rely on computationally-taxing updates to their entire hidden state at each timestep, a cost that grows with hidden state size. As demonstrated by the success of gating mechanisms such as the GRU (Cho et al., 2014) and LSTM (Hochreiter & Schmidhuber, 1997), all dimensions rarely need to be re-computed from scratch at each timestep. By *discretely* selecting which dimensions to update at each timestep via a learned *update pattern*, RNNs with a large hidden state can be trained with lower computational requirements (Bengio et al., 2013), inference in long RNNs can be expedited (Campos et al., 2018), and hidden representations may be made more robust to misleading inputs such as outliers or noise.

Selective neuron activation in RNNs has recently gained attention in the literature (Koutnik et al., 2014; Neil et al., 2016; Shen et al., 2019; Jernite et al., 2017; Campos et al., 2018). The most popular methods hand-craft specific *update patterns*, dictating which dimensions of the hidden state will update at which timesteps according to prior knowledge of a task (Koutnik et al., 2014; Neil et al., 2016). This imposes undue challenges in implementation, limits extensibility, and ignores the data-driven curation of information-flow through the RNN, a signature property of recurrent memory cells (Hochreiter & Schmidhuber, 1997; Cho et al., 2014). More recent methods learn to react to input data but impose strict relationships between the update patterns across both hidden dimensions and time (Shen et al., 2019; Jernite et al., 2017; Campos et al., 2018). While applicable to tasks with clear hierarchical components, such as modeling character-level text (Chung et al., 2017), these assumptions limit the expressiveness of the learned update patterns.

Specifically, we study the problem of generating a binary *update-pattern* for the hidden states learned by an RNN. The learned update-pattern defines which dimensions of the hidden state to update at each timestep, similar to the motivation for Residual Networks (He et al., 2016; Wang & Tian, 2016) and Highway Networks (Srivastava et al., 2015; Zilly et al., 2017). Ideally, only a small subset of the hidden state's dimensions needs to be updated at each timestep, especially with high-dimensional hidden states. In this way, representations can be learned while both solving a sequential learning task and minimizing the number of updates. This results in a reduction of the compute time. A solution to this multi-objective optimization problem should have a comparable accuracy to a traditional constantly-updating RNN but save the majority of computation steps along the way, ultimately accelerating inference and training (Neil et al., 2016).

Despite the potential for reducing computation required by RNNs, learning said update patterns is a challenging problem. First, binary-output neurons making discrete decisions (whether or not to update a hidden state dimension, for example) in the interior of a neural network is a classic challenge to gradient-based learning. This is because such decisions are non-differentiable by nature and therefore backpropagation cannot be directly used to update the weights. Second, the quality of a learned update pattern is unsupervised and thus the only feedback is task-specific. This discourages a priori assumptions of the update patterns.

To address the aforementioned challenges, we propose the *selective activation* RNN, or SA-RNN, which parameterizes a distribution of update-likelihoods, one per hidden state dimension, from which update-decisions can be made at each timestep. We augment an RNN with an update *coordinator* that adaptively controls which coordinate directions to update in the hidden state on the fly. The *coordinator* is modeled as a lightweight neural network that observes incoming data at each timestep and makes a discrete decision of whether or not enough information is stored in each individual hidden dimension to warrant an update. Subsequently, each hidden dimension is either computed by the RNN or copied from the previous timestep. The *coordinator*'s architecture is kept as simple as possible so the complexity of the RNN can scale without simply outsourcing computation to another network, similar to the controller in Ha & Schmidhuber (2018). Most notably, in contrast to other recent approaches (Koutnik et al., 2014; Jernite et al., 2017; Neil et al., 2016; Campos et al., 2018; Shen et al., 2019; Liu et al., 2018) we impose no assumptions of which individual hidden dimensions should update (or not update) together. Instead, we show that using an entirely-learned approach still results in complex task-specific update patterns. On three publicly-available datasets, we show that our low-bias approach achieves higher accuracy with far fewer updates than recent state-of-the-art methods (Koutnik et al., 2014; Jernite et al., 2017; Neil et al., 2016; Campos et al., 2018). These results indicate that predicting RNN update-patterns solely with respect to a task is not only feasible and low-bias, but is also favorable in a variety of settings.

## 2 RELATED WORK

Recurrent neuron update patterns have gained much interest in recent literature (Koutnik et al., 2014; Jernite et al., 2017; Neil et al., 2016; Chung et al., 2017; Shen et al., 2019; Liu et al., 2018). All of these methods boast fewer updates to the hidden states than standard RNN architectures. However, there are several limitations of these methods, two of which are summarized as follows.

First, the most popular methods rely on extensively-handcrafted update patterns consisting of periodic neuron activations (Koutnik et al., 2014; Neil et al., 2016; Liu et al., 2018). This requires either prior knowledge of sampling frequencies or seasonal patterns present in the data, reducing the potential extension to many sequential learning problems. Additionally, these input-agnostic periodic updates are fixed prior to learning. The choice of update periods heavily impacts the performance of the model, and sequences with irregular information flow cannot be modeled without massive state representations.

Second, the most recent works allow for *data-reactive* update patterns (Jernite et al., 2017; Shen et al., 2019; Campos et al., 2018) but assume temporal hierarchies in the input sequences and study settings where this effect is exceedingly obvious (for example, character-level sentence modeling (Chung et al., 2017)). In many real-world settings, temporal hierarchies are often subtle and forcing this assumption into the architectural design may limit its applications.

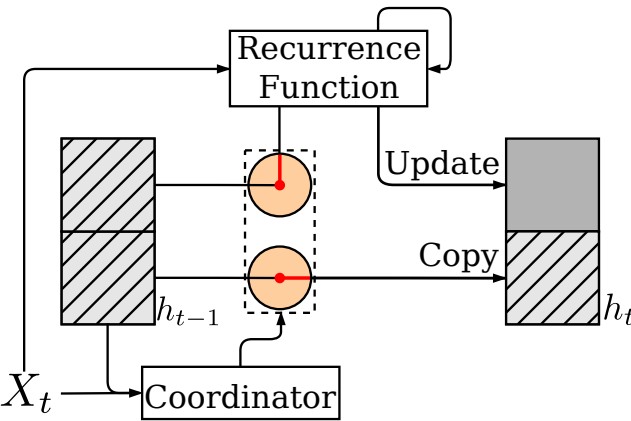

Figure 1: Overview of SA-RNN. $h_{t-1}$ is the hidden state at timestep $t-1$. Prior to computing $h_t$, the *update coordinator* decides which hidden dimensions will be updated according to $x_t$ and its previous update decisions. According to the decision in this figure, for example, hidden dimension 1 is updated while hidden dimension 2 remains unchanged.

Additionally, our approach is related to *conditional computation*, which predicts subsets of neural networks to activate depending on input data (Bengio et al., 2015; Shazeer et al., 2017; Cheng et al., 2017). In many cases, when a particular concept can be represented using only the sub-network of a large neural network, computation can be preserved by learning the structure of said sub-network (Schmidhuber, 2012) and activating it accordingly.

## 3 SELECTIVE NEURON ACTIVATION FOR RNNS

We introduce the **S**elective-**A**ctivation **RNN**, or SA-RNN, a broadly-applicable augmentation to RNNs which minimizes the computation required for RNNs by facilitating unimpeded information-flow across timesteps for individual dimensions of the hidden state. At its core, SA-RNN learns a data-driven strategy for discretely reading and writing information to the latent state space through the learned parameterization of an *update-likelihood* distribution. Despite leaving hidden dimension update patterns independent from one another, complex strategies still arise naturally depending on the sequential learning task at hand. In this section, we describe the training process of SA-RNN with $D$-dimensional hidden states on sequences of length $T$ for input data $x$ with $V$ variables. We omit biases from affine transformation equations and use notation for one training instance for ease of readability. An overview of the forward pass through SA-RNN is shown in Figure 1.

### 3.1 COMPUTING HIDDEN STATES

RNNs compute a sequence of hidden states one timestep at a time (Elman, 1990), each computed by a parametric recurrence function $R(\cdot)$: $h_t = R(h_{t-1}, x_t | \theta_r)$. The result is a sequence of vector representations $H = \{h_1, \ldots, h_T\}$ where each $h_t \in \mathbb{R}^D$ represents temporal dynamics of the time series up to timestep $t$ with respect to a task, preserving not only temporal dependencies but also the ordering of the inputs.

A popular and powerful augmentation to the RNN, as it was originally proposed, is the Gated Recurrent Unit (GRU) (Cho et al., 2014), which adds a series of gates between $h_{t-1}$ and $h_t$ to alleviate the vanishing gradient problem (Bengio et al., 1994):

$$r_t = \sigma(W_r h_{t-1} + U_r x_t) \tag{1}$$
$$z_t = \sigma(W_z h_{t-1} + U_z x_t) \tag{2}$$
$$s_t = \phi(W_c x_t + U_c(r_t \odot h_{t-1})) \tag{3}$$
$$\tilde{h}_t = (1 - z_t) \odot h_{t-1} + z_t \odot s_t \tag{4}$$

where $W$s and $U$s are matrices of learnable parameters of shape $D \times D$ and $D \times V$ respectively, $x_t \in \mathbb{R}^V$ is the input data at timestep $t$, $\odot$ represents the element-wise multiplication, $\sigma$ represents the sigmoid function, and $\phi$ represents a non-linearity (traditionally the hyperbolic tangent function). Its design is motivated heavily by the LSTM (Hochreiter & Schmidhuber, 1997). The GRU performs soft read/write operations, recomputing the entire vector $h_t$ at each timestep since gate $z \in [0, 1]^D$, the space of vectors with values inclusively between 0 and 1. Instead, we propose that all dimensions do *not* need to be updated at each timestep, as the position of the hidden state in many dimensions may often encode enough of the modeled input. Note that the output of the recurrence function is referred to as $\tilde{h}_t$. In the next section, we describe how to compute $h_t$, the final hidden state for timestep $t$ which is subsequently used for computing $h_{t+1}$ or the task.

## 3.2 SELECTIVE NEURON ACTIVATION

To reduce computation required to generate state representations, we assume updating representations to be a sequence of binary decisions – at each timestep a neuron will either be updated or not. Thus, we propose a learned *update coordinator*, which generates a binary mask for each hidden dimension, forecasting which dimensions need to be updated at the next timestep. First, an *update-likelihood* $\tilde{u}_t$ is computed for each neuron, informed by both the data observed at the current timestep and the previous update-likelihoods: $\tilde{u}_t = \sigma(W_u h_{t-1} + W_i x_t)$ where $W_u \in \mathbb{R}^{D \times D}$ is a diagonal matrix of trainable parameters which dictate the linear relationship between $h_{t-1}$ and $\tilde{u}_t$. $W_u$ is kept diagonal to maintain relationships between update-decisions of the same dimension while avoiding the extensive computation of a fully-connected layer, similar to the hidden state decay in Che et al. (2018). $W_i \in \mathbb{R}^{D \times V}$ encodes the influence of the input data on the current *update-likelihood* and $\sigma(\cdot)$ represents the hard sigmoid function, bounding $\tilde{u}$ according to a slope $\alpha$. Thus $\tilde{u}_t \in [0, 1]^D$, with one *update-likelihood* per dimension of the hidden state.

To discretize $\tilde{u}_t$, allowing information to flow unimpeded, element-wise binarization is applied:

$$u_t = \text{binarize}(\tilde{u}_t), \text{where} \tag{5}$$

$$\text{binarize}(a) = \begin{cases} 1 & \text{if } a > 0.5, \\ 0 & \text{otherwise.} \end{cases} \tag{6}$$

We apply this final discrete *update decision* as a binary gating mechanism since $u_t \in \{0, 1\}^D$:

$$h_t = u_t \odot \tilde{h}_t + (1 - u_t) \odot h_{t-1} \tag{7}$$

As written, this equation requires the pre-computation of $\tilde{h}_t$. However, through masking the computation can be directed at only the needed updates upon calculation of $u_t$. Thus when $\tilde{u}_t^n$, the *update decision* for the $n$-th dimension in $h$, is 1, $h_t^n$ is updated according to the new information present in $\tilde{h}_t^n$. We note that this update-decision strategy does not impose the inter-neuron assumptions of Jernite et al. (2017); Shen et al. (2019); Koutnik et al. (2014) while still allowing such strategies to be learned if they are found to be optimal by the model since decisions are made with respect to previous decisions, similar to Campos et al. (2018). We hypothesize that updating neurons together may generally be beneficial since complex temporal dependencies often require representations evolving in blocks of multiple neurons, as discussed in Koutnik et al. (2014).

Since binary-output neurons are inherently non-differentiable, barring the direct use of back-propagation, we approximate the gradient of the binarization function using the straight-through gradient estimator (Bengio et al., 2013) trained with slope-annealing (Chung et al., 2017):

$$\frac{\partial \text{binarize}(x)}{\partial x} = 1. \tag{8}$$

By estimating the gradient in this way we avoid additional loss terms and end up with empirically-reasonable approximations in comparison to other high-variance methods, such as REINFORCE (Williams, 1992; Chung et al., 2017; Campos et al., 2018). After computing the sequence of state representations $H$, they are projected into the output space depending on the task at hand.

## 3.3 TRAINING

All weights of SA-RNN are updated together using back-propagation to minimize one loss function. For readability we gather all weights into one parameter matrix $\theta$. Our loss function $J(\theta)$ consists of

two parts: a task-driven loss (denoted as $\mathcal{L}_{\text{task}}$) and an update-budget. The *task-driven* component may be be cross entropy for classification or possibly mean squared error for regression. The *update-budget* encourages sparser updates to the hidden dimensions, as shown in Equation 9 where $N$ is the number of training examples, $\hat{y}$ is the model's prediction, and $y$ is the label.

$$J(\theta) = \frac{1}{N} \sum_{i=1}^{N} \left( \mathcal{L}_{task}(y^i, \hat{y}^i) + \lambda \sum_{t=1}^{T} \tilde{u}_t^i \right) \tag{9}$$

$\lambda$ tunes the emphasis on the minimization of $\tilde{u}$, the *update probability*, and in practice it is reasonable to set $\lambda = 0$, encouraging the model to make *update decisions solely with respect to the learning task*. As $\lambda$ is increased and update likelihoods decrease, the hidden dimension is update fewer times until eventually $h_0 = h_T$ where the hidden state is not updated at all.

## 4 EXPERIMENTS

### 4.1 DATASETS

We conduct our evaluation using three classification tasks on the following datasets, each being publicly-available.

`Seizures`[1] (Andrzejak et al., 2001): From 11,800 178-timestep time series, the task is to detect which EEGs contain evidence of epileptic seizure activity. Since there are only 2,300 cases of such activity, we down-sample an equal number from the negative class, resulting in a balanced dataset with 4,600 time series. Finally, we center the time series around zero and compute the mean value of every 10-timestep chunk, summarizing each series into 17 final timesteps.

`TwitterBuzz`[2] (Kawala et al., 2013): To predict buzz events on Twitter, we work with 77-dimensional time series with labels indicating whether or not a spike in tweets on a particular topic is observed. Starting with over 140,000 timesteps, we compute the mean of every five steps, center the time series around zero, and break the resulting 28,000 timesteps into 2,800 length-15 sequences. We then extract the 776 time series containing any buzz events and balance the dataset by randomly selecting an equal number of no-buzz time series, resulting in 1552 labeled time series.

`Yahoo`[3]: We re-frame this outlier detection dataset as a classification problem: whether or not a sequence contains an anomaly. We begin by chunking the time series into subsequences of length 25. Then, we create a balanced dataset by selecting all subsequences with anomalies present along with an equal number of randomly-selected subsequences with no anomalies to serve as our negative class. Thus we end up with a dataset containing 418 length-25 time series.

### 4.2 SETTINGS

**Baselines.**
To evaluate the performance of SA-RNN, we compare with five recent state-of-the-art related methods:

- **Random Skips**: This method is effectively the random version of our proposed method. At each step, random hidden dimensions are updated. This is similar to *Zoneout* (Krueger et al., 2017) however we maintain random updating during testing.

- **Clockwork RNN** (Koutnik et al., 2014): Hidden dimensions are updated in groups at pre-determined "clock" rates. For example, the first five dimensions in the hidden state may update every step while the second five neurons update every 3 steps.

- **Phased LSTM** (Neil et al., 2016): Hidden dimensions are updated independently at sampled "clock" rates where the user defines the distribution from which to sample. Each hidden dimension has its own update pattern.

- **VC-GRU** (Jernite et al., 2017): A value $p \in [0, 1]$ is predicted at each step indicating the proportion of the representation to update. Then, the first $p * D$ dimensions are updated, imposing a hierarchical structure to the update patterns.

---

[1]http://epileptologie-bonn.de/cms/front_content.php?idcat=193&lang=3&changelang=3
[2]http://ama.liglab.fr/resourcestools/datasets/buzz-prediction-in-social-media/
[3]https://webscope.sandbox.yahoo.com/catalog.php?datatype=s&did=70

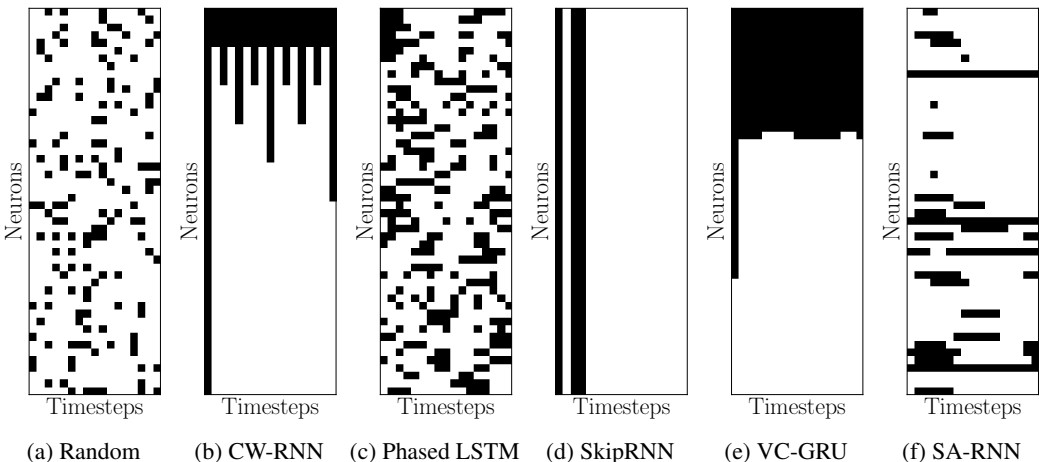

Figure 2: Sample skipping patterns from compared methods for the `Seizures` task with 50-dimensional state representations. Black squares indicate *update* while white squares indicate *skip*.

- **SkipRNN** (Campos et al., 2018): Hidden dimensions update in lock-step with one another so the entire hidden state is either modified or left unchanged at each step. Additionally, the update likelihoods increase monotonically, regardless of input data.

**Implementation Details.**
For all experiments, we use an 80% training, 10% validation, and 10% testing dataset split. The training data are used to tune the parameters of the models; the validation data are used to validate hyperparameter selection; the testing data are used to report final performances. We randomly repeat this process ten times to compute confidence intervals for performance metrics. Across all models we fix the size of state representations to be fifty neurons. Keeping this number fixed allows us to purely compare performance of the update-patterns observed in alternative algorithms. For $\lambda$ selection in SkipRNN and our proposed method, SA-RNN, we search in a log-space ranging from 0.0 to 0.1 in 11 steps. For all methods, we choose the same learning rate of $1e^{-03}$ after a log-space search ranging from $1e^{-05}$ to $1e^{-01}$, likely due to similarities behind the core of the sequential learning (RNNs with equal state representation sizes). We optimize all models using Adam (Kingma & Ba, 2014). While we describe our method in the case of the GRU, it may also be directly applied to other gating mechanisms such as the LSTM. We implement *slope annealing* following the setting described in Chung et al. (2017).The code for our method is available at `https://hiddenforreview.com`.

## 4.3 EXPERIMENTAL RESULTS

**Contrasting update patterns of alternative algorithms.**
First, we inspect the update-patterns generated by each baseline algorithm, as shown in Figure 4. Both PhasedLSTM and VC-GRU use *partial* updates, and do not make fully-binary update decisions. For this visualization we binarize their update patterns using a ceiling function on non-zero update decisions since the neurons are still updated. Importantly, all algorithms are subject to update-budgets, which clearly effect the observed patterns. For example, since the SkipRNN updates all neurons at the same time, deciding to update has a large cost, eating through the update budget quickly. Meanwhile, other methods such as our own SA-RNN and the PhasedLSTM take much smaller bites into the update budget since the neurons are updated independently. From these visuals, it is clear that ours is the only method which can adaptively learn to update independent neurons for many steps at a time, according to the data. Thus, some neurons may update a few times in the middle of the sequence instead of at the beginning or end, a decision which is driven directly by the data. This allows our method to spend its computational budget in a more informed way, choosing to activate some neurons many times in a row, while leaving others unchanged.

**Comparing accuracy and update-frequency across methods.**
Second, we show results from the three tasks described in Section 4.1. We measure four properties of

| Method | $\sqrt{\text{Acc.} \times \text{Skip}\%}$ | Skip(%) | Accuracy (%) | FLOPs |
|---|---|---|---|---|
| GRU (Cho et al., 2014) | 0 | 0 | 84.7 (0.8) | 257550 |
| CW-RNN (Koutnik et al., 2014) | 73.7 | 81 | 67.1 (0.4) | 16150 |
| PhasedLSTM (Neil et al., 2016) | 74.9 | 69 | 81.4 (1.4) | 106454 |
| VC-GRU (Jernite et al., 2017) | 68.7 | 66 | 71.6 (3.4) | 89284 |
| SkipRNN (Campos et al., 2018) | 69.2 | 82 | 58.4 (5.5) | 48042 |
| Random Skips | 62.3 | 76 | 51.1 (0.1) | 61812 |
| SA-RNN | **78.8** | 76 | 81.6 (0.6) | 63512 |

Table 1: Performance on the `Seizures` task. $\sqrt{\text{Accuracy} \times \text{Skip}\%}$ is the geometric mean of Accuracy and Skip Percent. The first section contains the baseline GRU, the second contains non-reactive methods, the third contains learned methods, and the fourth is our proposed method and random baseline. For SA-RNN we use $\lambda = 2e^{-4}$.

| Method | $\sqrt{\text{Acc.} \times \text{Skip}\%}$ | Skip (%) | Accuracy (%) | FLOPs |
|---|---|---|---|---|
| GRU (Cho et al., 2014) | 0 | 0 | 64.8 (1.6) | 378750 |
| CW-RNN (Koutnik et al., 2014) | 68.2 | 81 | 57.4 (1.9) | 23750 |
| PhasedLSTM (Neil et al., 2016) | 65.3 | 70 | 60.9 (4.0) | 151500 |
| VC-GRU (Jernite et al., 2017) | 62.9 | 66 | 59.9 (2.8) | 131300 |
| SkipRNN (Campos et al., 2018) | 69.2 | 88 | 54.4 (8.1) | 47925 |
| Random Skips | 72.2 | 89 | 58.6 (1.9) | 41662 |
| SA-RNN | **75.0** | 89 | 63.1 (1.8) | 44162 |

Table 2: Performance on the `Yahoo` task. For SA-RNN we use $\lambda = 1e^{-3}$.

each method. 1) *Skip Percent* – the proportion of neurons which are *not* updated, computed across all timesteps. To make a fair comparison between all methods, we tune their hyperparameters so that their *skip percents* are as close together as possible. This allows us to better analyze the effects of the differences in update patterns instead of changing the hidden state dimension sizes. 2) *Accuracy* – the accuracy achieved on the task. For each task we use balanced datasets, so the lower bound is 50%. 3) $\sqrt{\text{Accuracy} \times \text{Skip}\%}$ – the geometric mean of a model's accuracy and skip percent. The measure allows comparison of all methods since they do not all update the exact same number of times. Thus, this is the key performance metric, the upper and lower limits of which are 1.0 and 0.0, respectively. 4) *FLOPs* – as in Campos et al. (2018) we compute the FLOPs for each method as a surrogate for wall-clock time, which is hardware-dependent and often fluctuates dramatically in practice.

Across all tasks, SA-RNN achieves on average higher accuracy with fewer updates, as shown in Tables 1, 2, and 3. In `Yahoo` and `TwitterBuzz`, SA-RNN maintains by far the closest accuracy to that of the baseline GRU, which updates every hidden dimension at every timestep. In `Seizures`, both SA-RNN and PhasedLSTM are similar to the GRU, possibly due to periodic dynamics in the data. We also show in Tables 1 and 3, the accuracy of *Random Skips* is far worse than SA-RNN, indicating that the benefits of our proposed update-strategy comes strongly from the *learning*. The benefits of adaptive update patterns are not present in the Clockwork RNN or PhasedLSTM.

SA-RNN has nearly-equivalent FLOPs to the other methods that learn update patterns since each adds another affine transformation to map data to update decisions. As shown in Table 3, adapting the update decisions according to the input data adds a significant amount of computation with high-dimensional inputs. To improve this, update patterns may be predicted for multiple steps concurrently, depending on the input data. Given the same number of updates, the FLOPs are roughly equivalent between our method, VC-GRU, and SkipRNN. As in Campos et al. (2018), it may be possible to only observe the previous hidden state, however this results in perpetual lag as information fills the hidden state. CW-RNN consistently has extremely low FLOPs since there is no data-driven decision making in the update patterns, instead hard-coding them beforehand. Additionally, their recurrence function is not a gated recurrent memory cell.

| Method | $\sqrt{\text{Acc.} \times \text{Skip\%}}$ | Skip (%) | Accuracy (%) | FLOPs |
|---|---|---|---|---|
| GRU (Cho et al., 2014) | 0 | 0 | 82.8 (2.2) | 569250 |
| CW-RNN (Koutnik et al., 2014) | 72.1 | 81 | 64.2 (0.8) | 35910 |
| PhasedLSTM (Neil et al., 2016) | 70.2 | 70 | 70.5 (1.8) | 227700 |
| VC-GRU (Jernite et al., 2017) | 61.9 | 66 | 58.1 (2.2) | 197340 |
| SkipRNN (Campos et al., 2018) | 72.2 | 87 | 60.0 (1.1) | 75487 |
| Random Skips | 61.2 | 76 | 49.4 (0.7) | 130927 |
| SA-RNN | **76.1** | 77 | 75.4 (3.0) | 252120 |

Table 3: Performance on the `TwitterBuzz` task. For SA-RNN we use $\lambda = 2e^{-3}$.

(a) `Twitter`  (b) `Seizures`  (c) `Yahoo`

(d) `TwitterBuzz`  (e) `Seizures`  (f) `Yahoo`

Figure 3: Observing the effect of skip percent and $\lambda$ on accuracy. Accuracy and skip percent contrast one another, resulting in a trade-off.

**Effects of budgeting neuron updates.**
Finally, we assess how the performance of SA-RNN depends on its hyperparameter $\lambda$, which budgets the number of permitted updates. We investigate $\lambda$ values from 0 to 0.1 on a log-scale, since empirically all updates stopped when $\lambda > 0.1$. Interestingly, the relationship between accuracy and $\lambda$ depends heavily on the task, as demonstrated in in the first row of Figure 3. For example, on the `TwitterBuzz` task, we observe a smooth transition from random guessing (when no updates are allowed) to our peak accuracy (no constraint on updates). Meanwhile, on the `Seizures` task, there is a sharp increase from random predictions to near-peak performance. This could be a feature of the parameter search space, but with already-small changes to $\lambda$ this indicates high-sensitivity. We also investigate how accuracy changes as a function of the number of neurons skipped, as shown in the second row of Figure 3. Interestingly, there are steep elbows where very few updates are needed to observe near-peak accuracy. This bolsters the intuition behind Residual Networks and related methods: many hidden dimensions often capture enough information to warrant direct copying.

## 5 CONCLUSIONS

In this paper, we study the problem of reducing the number of updates to the state representations learned by RNNs. This allows for less computation at each timestep. We propose an augmentation to general RNN models, called SA-RNN, which is carefully crafted to skip neurons while maintaining accuracy through the parameterization of a distribution of neuron update-likelihoods, making binary decisions for each neuron at each step. We conduct extensive experiments on three real-world publicly-available datasets and show that our method achieves on average higher accuracy with fewer neuron updates compared to recent state-of-the-art alternatives. Our results demonstrate that updating state representations without imposing high-bias decisions on the update-patterns is not only easier to implement and train, but preferable in terms of performance.

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

## 6 APPENDIX

### 6.1 ADDING TASK

In this experiment, the network must learn to use a binary mask to sum the corresponding values in a real-valued sequence. This task was classically used to validate the LSTM's effectiveness and is now commonly used to evaluate the robustness of memory in RNNs in the presence of extremely long-term dependencies (Hochreiter & Schmidhuber, 1997; Neil et al., 2016; Campos et al., 2018). The task is as follows: given a data sequence $x$ of $T$ values sampled uniformly between 0 and 1 and a binary masking sequence $m$ containing $T - 2$ zeros and 2 ones which are located at indices $i$ and $j$, the ground truth $y$ is computed as $y = x_i + x_j$. Given a new data sequence $x$ and a masking sequence $m$, the mean squared error between $y$ and $f_\theta(x, m)$ should be approximately 0.

We set $T = 500$ to stress-test long-term dependencies and use 128-dimensional hidden states, as suggested in Henaff et al. (2016).

| Method | Task solved | Skip Percent | FLOPs ($\times 10^6$) |
|---|---|---|---|
| GRU | Yes | 0.0 | 49.7 |
| Random Skips | No | 25.0 | 24.9 |
| Random Skips | No | 50.0 | 12.4 |
| Random Skips | No | 90.0 | 4.97 |
| SkipRNN ($\lambda = 0$) | Yes | 2.1 (3.3) | 44.8 |
| SkipRNN ($\lambda = 1e^{-7}$) | Yes | 21.6 (39.7) | 39.6 |
| SA-RNN ($\lambda = 0$) | Yes | 25.2 (5.1) | 37.5 |
| SA-RNN ($\lambda = 1e^{-5}$) | Yes | 49.6 (9.6) | 25.3 |
| SA-RNN ($\lambda = 1e^{-4}$) | Yes | 90.0 (3.9) | 15.3 |

Table 4: Adding task with 500 timesteps.

As shown in Table 4, SA-RNN consistently solves the adding task with far higher *skip percentage* and lower FLOPs than the baseline GRU and the SkipRNN. Even though SkipRNN theoretically can and should find the proper timesteps to observe given the mask, it did not solve the problem consistently, only performing well under a couple of choice settings which updated the vast majority of the hidden states. However, we note that the sequence-length $T$ used in this experiment is ten times greater than was originally explored in Campos et al. (2018).

### 6.2 HYPERPARAMETER SELECTION FOR COMPARED METHODS

We employ the validation set to select hyperparameters for compared models described in Section 4.2.

- *Random Updates*: When training Random Updates, the one hyperparameter to tune is the *probability of updating each hidden dimension*. Our goal is to match the number updates in Random Updates to the number observed in our proposed method, SA-RNN. So after observing how many times SA-RNN updated, we simply create random binary masks with the same update proportions.

- *Clockwork RNN*: This method relies on hand-crafted update patterns. In order to have roughly 20% of the updates (to be similar to the other methods) with 50-dimension state representations, we use block-sizes of 5 (the number of dimensions whose updates are tied together) with exponentially-increasing clock rates as proposed in the original paper: $\{1, 2, 4, 8, 16, 32, 64, 128, 264, 512\}$. We choose this as it is the originally-intended use of the model and resulted in the proper number of updates. There is no variance in this method as it does not react to the input data.

- *PhasedLSTM*: This method also hand-crafted update patterns, sampling update periods for individual hidden dimensions. The distribution from which to sample update periods is left up to the user (along with the parameters of such distribution). As proposed in the paper, we sample update periods from a uniform distribution $\mathcal{U}$. To determine the lower and upper bounds of the distribution, we tried sampling update patterns from a uniform distribution of

all combinations of integers between 0 and 6, finally choosing $\mathcal{U}(1, 3)$. We sample $D$ values from this distribution, where $D$ is the dimension of the state representation. We choose the *shift* as described in the original PhasedLSTM paper, sampling a value with an upper limit of the period for each neuron. These two settings, paired with $r_{\text{on}} = 0.3$ led to a comparable number of updates.

- *VC-GRU*: This method predicts proportions of the state representation to update at each timestep. During the optimization, there is a loss term which adds a "target" proportion, penalizing deviations from this target value. As suggested in the original paper, we tried values of $0.1$, $0.2$, and $0.5$ and selected $0.1$, which resulted in a comparable number of updates.

- *SkipRNN*: This method predicts likelihoods of state updates at each timestep. In the loss function, there is one hyperparameter, $\lambda$, which penalizes the number of updates. We use a log-space search, ranging from $0.0$ to $0.1$ in 11 steps, settling on $1e - 05$ for the `Twitter` task, and $1e - 04$ for the `Seizures` and `Yahoo` tasks. We found the number of updates to be extremely sensitive, skipping from roughly 13% to 100% between single log-space steps.

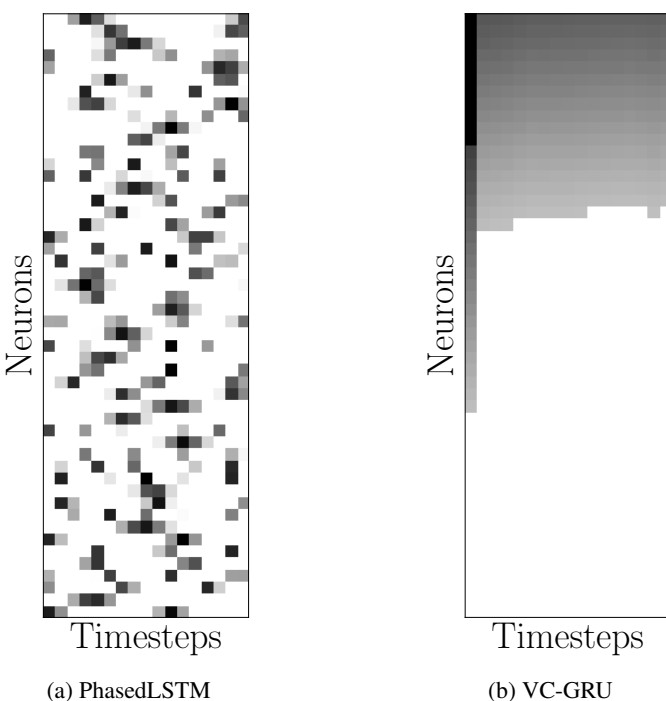

(a) PhasedLSTM        (b) VC-GRU

Figure 4: Raw update patterns. Each column is a state representation. White boxes indicate *Skip*, darker boxes indicate the degree to which a neuron is updated with darker being more updated. A black box indicates that a neuron is fully-updated.

## 6.3 VISUALIZING UPDATE PATTERNS

In Section 4.3 we visualize and contrast the update patterns between all described methods. PhasedLSTM and VC-GRU, however, do not propose fully-binary update patterns. PhasedLSTM softly opens and closes neurons, and VC-GRU approximates binary update-decisions using a soft mask, changing the proportion of updated-neurons at each timestep. In order to fairly compare them to all other methods, we employ ceiling functions to raise their update predictions to 1 (prior to this, their updates are between 0 and 1, similar to LSTM and GRU). In Figure 4, we display their update patterns prior to this ceiling function, to show their use as originally intended. We believe that this use of the ceiling function is fair, since such soft updates are still updates, which is the focus of our work.

### 6.4 IMPLEMENTING SA-RNN

We implement our method (and all other methods) in PyTorch 1.0. Below, we provide some pseudocode to give a taste of how to approach implementing SA-RNN (at a high level). The pseudocode is loosely formatted similar to PyTorch for readability. In the pseudocode, `Discriminator` is simply a network which maps the hidden state to the latent space for the final prediction. Upon acceptance of the paper, the full code will be released along with our entire training process and many examples. In practice the next state does not need to be fully computed once the update decisions are computed.

```
def forward(sequence):
    u_likelihoods = zeros # Initialize update likelihoods as zeros
    state = intializeState() # Create the initial state
    for t in range(len(sequence)): # Loop through sequence.
        x = sequence[t] # Extract one timestep of data
        u_likelihoods = Coordinator.forward(x, u_likelihoods)
        u_decisions = Binarize.forward(u_likelihoods)
        state_tilde = RNN.forward(x, state)
        state = u_decisions*state_tilde + (1-u_decisions)*state
    prediction = Discriminator.forward(state)
    return prediction
```

