# OpenReview forum: "Reducing Computation in Recurrent Networks by Selectively Updating State Neurons"
_ICLR.cc/2020/Conference — Reject_

### Official Review · AnonReviewer2 · 2019-10-20
**Official Blind Review #2**

**Rating:** 6

**Review:**

Summary: This paper proposes selective activation RNN (SA-RNN), by using an update coordinator to determine which subset of the RNN’s hidden state dimensions should be updated at a given timestep. The proposed loss term is then a sum of the original objective (e.g. classification) and a weighted sum of the probability that each dimension will be updated for each timestep. The method is evaluated on 3 time series datasets: Seizures, TwitterBuzz, Yahoo.

Decision: Weak Reject. Although the authors tackle a challenging problem, their empirical results are lacking to provably demonstrate that their approach outperforms existing baselines.

Supporting Arguments/Feedback: The authors compare SA-RNN to 5 baselines: random updates, clockwork RNN, phased LSTM, Skip RNN, and VC-GRU. Although I appreciated the authors’ comparison across the suite of methods with respect to various metrics (e.g. # FLOPS, proportion of neurons that weren’t updated, etc.), the experiments were conducted on datasets that were relatively simple. For example, in prior work, the empirical evaluations were on much larger-scale datasets such as Wikipedia [Shen et. al 2019], real clinical data sources [Liu et. al 2018], and Charades videos [Campos et. al 2018], among others. I would be very interested to see how this training procedure fairs when evaluated on much more complex tasks, and would make the results about computational speedups at train/test time much more convincing.

Questions:
- I’m curious if you tried different types of gradient estimators to get around the non-differentiability rather than the straight-through estimator. Also how was the slope-annealing conducted (e.g. annealing schedule)?


**Experience Assessment:**

I do not know much about this area.

**Review Assessment: Checking Correctness Of Derivations And Theory:**

I assessed the sensibility of the derivations and theory.

**Review Assessment: Checking Correctness Of Experiments:**

I assessed the sensibility of the experiments.

**Review Assessment: Thoroughness In Paper Reading:**

I read the paper at least twice and used my best judgement in assessing the paper.

---

> ### Author Response · Authors · 2019-11-15
> **Reply to reviewer 2 - Thank you for thoughtful review**
>
> Thank you very much for your constructive review.
>
> Per your suggestion, we have run further experiments on more complicated data that we hope will convince you of both the merit of our proposed method along with this line of research with respect to previous approaches. We revised the paper, adding a new section to the beginning of the appendix that describes the “Adding” task, which asks a network to learn to sum two values in a long sequence of sampled values given a mask indicating the indices to be summed, along with our results. We invite you to take a look at the new section but we highlight our main findings as follows:
>
> To add a new perspective on complexity in our experiments we tested long-term dependencies as part of this new experiment, following the lead of the literature [1-3]. As shown in our Appendix, we found that even in the presence of extremely long-term dependencies (up to 500 timesteps), our proposed SA-RNN solves the task perfectly with a very low number of FLOPs and very few state updates compared to the other data-reactive method SkipRNN. As expected, the standard GRU also solves the task while Random Skips does not.
>
> Regarding your Gradient Estimation and Slope-Annealing question:
> In our experiments, we used the straight-through estimator to be comparable to the literature, including [3] and [4].  Plus, we were pleased to have achieved our empirically-good results with these basic settings. However, we agree that this is an interesting question, as it is unlikely that the same estimator is the absolute best for all possible tasks. Thus, there is potentially room for further tuning by designing update pattern-specific gradient estimation. For slope-annealing we used the parameters and setting as described in [4], gradually increasing the slope of the hard sigma function as the model trains, starting at slope $\alpha=1$ and increasing according to the schedule $a = \min(5, 1+0.04*N_{epoch})$. We have added information about both of these settings to our experimental description in our paper to assure full reproducibility.
>
> [1] Henaff, M., Szlam, A., LeCun, Y. “Recurrent Orthogonal Networks and Long-Memory Tasks”, ICML 2015.
> [2] Neil, D., Pfeiffer, M., Liu, S.-C., “Phased LSTM: Accelerating Recurrent Network Training for Long or Event-based Sequences”, NeurIPS 2016.
> [3] Campos, V., Jou, B., Giro-i-Nieto, X., Torres, J., Chang, S.-F. “SkipRNN: Learning to Skip State Updates in Recurrent Neural Networks”, ICLR 2018.
> [4] Chung, J., Ahn, S., Bengio, Y. “Hierarchical Multiscale Recurrent Neural Networks”, ICLR 2017.

---

> > ### Comment · AnonReviewer2 · 2019-11-16
> > **thanks for changes**
> >
> > Thanks for the additional clarification and experiment, it helped to contextualize the difficulty of the problem and value added of the method. I've revised my score accordingly.

---

### Official Review · AnonReviewer3 · 2019-10-22
**Official Blind Review #3**

**Rating:** 6

**Review:**

This paper attempts to reduce computation in recurrent neural networks. Instead of artificially determining the update pattern for updating the states, the authors propose SA-RNN to predict discrete update patterns automatically through optimization driven entirely by the input data. Experiments on publicly-available datasets show that the proposed method has competitive performance with even fewer updates.

Pros:
Overall, I think the idea of this paper is clear and the whole paper is easy to follow. The experiments clearly show the advantage of the proposed method claimed by the authors.

Cons:
1.	Some expressions need to be improved. For example, in “This way, representations can be learned while solving a sequential learning task while minimizing the number of updates, subsequently reducing compute time.” two “while”s are not elegant and there should be an “In” before “this way”. In “We augment an RNN with an update coordinator that adaptively controls the coordinate directions in which to update the hidden state on the fly”, the usage of “in which to” is not right. I suggest the authors to thoroughly proofread the whole paper and improve the presentation.
2.	Since this paper focuses on the efficiency of RNN, I suggest the authors could provide the time complexity comparisons. Merely the comparisons on skip of neurons cannot show the advantage on the efficiency.


**Experience Assessment:**

I have published one or two papers in this area.

**Review Assessment: Checking Correctness Of Derivations And Theory:**

I carefully checked the derivations and theory.

**Review Assessment: Checking Correctness Of Experiments:**

I carefully checked the experiments.

**Review Assessment: Thoroughness In Paper Reading:**

I read the paper thoroughly.

---

> ### Author Response · Authors · 2019-11-09
> **Response to Reviewer 3 - Thank you for positive feedback + expression improvement**
>
> We thank you for your time and effort in reviewing our work and your positive response to our proposed method and experimental results, especially given your expertise in the area.
>
> Concerning your suggestions to improve the presentation itself, we have undertaken a careful round of proof-reading to improve the readability of the manuscript. In addition, we had a colleague in the English department provide additional editing suggestions. We have now uploaded a new version of the paper addressing both your specific edits as well as this general round of proof-reading. We invite you to take a look at the revised presentation.
>
> Timing comparisons: Metrics such as wall-clock time greatly depend on factors outside the model such as implementation strategy, machine learning framework, and hardware specifics. To target the methodological differences between our compared methods, we compute and report the FLOPs instead. This is independent of hardware and implementation. Instead, it directly compares the computational requirements of the update-mechanisms, as described in [1] for a fairer comparison.
>
> [1] Campos et. al, “SkipRNN: Learning to Skip State Updates in Recurrent Neural Networks”, ICLR 2018.

---

### Official Review · AnonReviewer1 · 2019-10-23
**Official Blind Review #1**

**Rating:** 6

**Review:**

A main problem with RNN is to update all hidden dimensions in each time step. The authors proposed selective-activation RNN (SA-RNN), which modifies each state of RNN by adding an update coordinator which is modeled as a lightweight neural network. The coordinator, based on the incoming data, makes a discrete decision to update or not update each individual hidden dimension. A multi-objective optimization problem is defined to both solving a sequential learning task and minimizing the number of updates in each time step. The authors evaluated their networks on three public benchmark datasets and achieved good results compared to the state-of-the-art ones.
The papers is well-written. The idea proposed in this paper is interesting and it is presented very well. There is also an extensive evaluation.


**Experience Assessment:**

I have published one or two papers in this area.

**Review Assessment: Checking Correctness Of Derivations And Theory:**

I carefully checked the derivations and theory.

**Review Assessment: Checking Correctness Of Experiments:**

I carefully checked the experiments.

**Review Assessment: Thoroughness In Paper Reading:**

I read the paper at least twice and used my best judgement in assessing the paper.

---

> ### Author Response · Authors · 2019-11-08
> **Respose to Reviewer 1 - Thank you for positive feedback.**
>
> We greatly appreciate your positive feedback on our method, presentation, and experimental results, especially given your expertise in the area.

---

### Decision · Program_Chairs · 2019-12-19

**Decision:**

Reject

**Comment:**

This paper introduces a new RNN architecture which uses a small network to decide which cells get updated at each time step, with the goal of reducing computational cost.  The idea makes sense, although it requires the use of a heuristic gradient estimator because of the non-differentiability of the update gate.

The main problem with this paper in my view is that the reduction in FLOPS was not demonstrated to correspond to a reduction in wallclock time, and I don't expect it would, since the sparse updates are different for each example in each batch, and only affect one hidden unit at a time.  The only discussion of this problem is "we compute the FLOPs for each method as a surrogate for wall-clock time, which is hardware-dependent and often fluctuates dramatically in practice."  Because this method reduces predictive accuracy, the reduction in FLOPS should be worth it!

Minor criticism:
1) Figure 1 is confusing, showing not the proposed architecture in general but instead the connections remaining after computing the sparse updates.